# Leveraging multi-way interactions for systematic prediction of pre-clinical drug combination effects

Heli Julkunen [1], Anna Cichonska [1,2,3], Prson Gautam [3], Sandor Szedmak [1], Jane Douat[1], Tapio Pahikkala [2], Tero Aittokallio [1,3,4,5,6 ✉] & Juho Rousu [1 ✉]

We present comboFM, a machine learning framework for predicting the responses of drug combinations in pre-clinical studies, such as those based on cell lines or patient-derived cells. comboFM models the cell context-specific drug interactions through higher-order tensors, and efficiently learns latent factors of the tensor using powerful factorization machines. The approach enables comboFM to leverage information from previous experiments performed on similar drugs and cells when predicting responses of new combinations in so far untested cells; thereby, it achieves highly accurate predictions despite sparsely populated data tensors. We demonstrate high predictive performance of comboFM in various prediction scenarios using data from cancer cell line pharmacogenomic screens. Subsequent experimental validation of a set of previously untested drug combinations further supports the practical and robust applicability of comboFM. For instance, we confirm a novel synergy between anaplastic lymphoma kinase (ALK) inhibitor crizotinib and proteasome inhibitor bortezomib in lymphoma cells. Overall, our results demonstrate that comboFM provides an effective means for systematic pre-screening of drug combinations to support precision oncology applications.

[1] Department of Computer Science, Helsinki Institute for Information Technology HIIT, Aalto University, Espoo, Finland. [2] Department of Future Technologies, University of Turku, Turku, Finland. [3] Institute for Molecular Medicine Finland FIMM, University of Helsinki, Helsinki, Finland. [4] Department of Mathematics and Statistics, University of Turku, Turku, Finland. [5] Department of Cancer Genetics, Institute for Cancer Research, Oslo University Hospital, Oslo, Norway. [6] Oslo Centre for Biostatistics and Epidemiology, University of Oslo, Oslo, Norway. ✉email: tero.aittokallio@helsinki.fi; juho.rousu@aalto.fi

Combination therapies are often required for treating cancer patients with advanced stages of the disease. In addition to overcoming monotherapy resistance, combinatorial treatments can also reduce toxicity of the treatment (by reduced doses of the drugs) and improve therapeutic efficacy (by multi-targeting effect)[1–3]. With recent advances in high-throughput screening methods, a systematic evaluation of combinations among large collections of chemical compounds has become feasible. This typically leads to large-scale experiments, in which the combinatorial responses are tested in various doses of the individual compounds, resulting in dose–response matrices that capture the measured combination effects for every concentration pair in a particular sample (e.g., cancer cell line or patient-derived cells)[4]. However, even with modern high-throughput instruments, experimental screening of drug combinations quickly becomes impractical, as the number of conceivable drug combinations increases rapidly with the number of drugs in consideration. In addition, the inherent heterogeneity of cancer cells pose further challenges for the experimental efforts, as the combinations need to be tested in various cell contexts and genomic backgrounds[5,6]. Therefore, computational methods are often being used to guide the discovery of effective combinations to be prioritized for further pre-clinical and clinical validation[7,8].

During the recent years, machine learning has emerged as a powerful approach to aid the drug development process by offering systematic means for the prediction of target bioactivities and drug-induced effects[9–13], thereby providing guidance for drug discovery and repositioning efforts[14,15]. Until recently, the performance of machine learning methods in predicting drug combination effects was limited by the lack of high-quality training data[8]. However, this is gradually changing as increasing amounts of data from pre-clinical drug combination screens are becoming available, therefore creating new opportunities also for the application of large-scale machine learning methods[4,5,16]. For instance, the NCI-ALMANAC dataset generated by the US National Cancer Institute (NCI) provides over 3 million experimentally measured drug combination responses across various cell lines and tissue types[4]. However, despite the potential value of such datasets, the high dimensionality of the underlying dose–response data and the inherent complexity of drug interaction patterns across various doses pose challenges to accurate modeling of drug combination effects.

Several computational tools have been proposed for the prediction of drug combinations[2,7,8,17]. Many of these tools have been systematically benchmarked in two crowdsourced DREAM Challenge competitions[18,19], which demonstrated that computational predictions can achieve high accuracies for selected drug classes, provided there are enough drug information and training data available. However, the focus of these challenges and most of the previously proposed methods has been on directly predicting drug combination synergies (i.e., whether the combined summary effect is higher than expected). In many practical applications, however, more detailed information on dose–response effects of the combinations is required, rather than simply classifying the summary effects into synergistic or antagonistic classes. Furthermore, as noted in the recent AstraZeneca-Sanger drug combination prediction DREAM challenge[19], the performance of the computational methods typically relies on selective incorporation of target features and biological knowledge that is not always available for all drugs and cell models. Therefore, there is a need to develop integrative and robust models capable of generalizing and learning from large amounts of available data that facilitate the exploration of the extensive combinatorial drug and dose spaces.

Here, we present comboFM, a novel machine learning framework for systematic modeling of drug-dose combination effects in a cell context-specific manner. It is generally applicable to any pre-clinical model systems, such as patient-derived primary cells, but we demonstrate its performance here in cancer cell lines (Fig. 1). We base our work on the observation that the drug combination dose–response data can be compiled into a higher-order tensor indexed by drugs, drug concentrations, and cell lines. comboFM then models the cell line-specific responses to a combination of drugs as an interaction between the different modes of the tensor using a higher-order factorization machine (FM)[20], a recently proposed machine learning approach for non-linear learning on large data. FMs have been shown to be compelling tools with the ability to work particularly well with high-dimensional and sparse datasets[20–22]. In contrast to existing machine learning models, comboFM enables one to explore the detailed landscape of drug combination responses across various doses. We demonstrate that comboFM obtains high prediction accuracy in various practical application scenarios, significantly outperforming other approaches. Furthermore, we show the robustness and practical potential of comboFM by experimentally validating untested drug combinations predicted for specific cell lines.

## Results

**Overview of comboFM model**. comboFM was developed for predicting drug combination responses of cancer cell lines in three practical scenarios (Fig. 1a). The first scenario of predicting new dose–response matrix entries corresponds to filling in the gaps in partially measured dose–response matrices. In the second scenario of new dose–response matrix inference, the predictions are made for completely held out dose–response matrices of untested drug–drug–cell line triplets, such that the drug pair has still been observed in other cell lines. In the third and most challenging scenario of new drug combination inference, the predictions are made for completely new drug combinations with no available combination measurements in any cell line, thereby providing guidance on repositioning of the drugs for new combinations and cell contexts.

To capture the high-order interactions between drug combinations in different cell lines and at various doses, comboFM models the multi-way interactions between the two drugs, the cell lines and the dose–response matrices as a fifth-order data tensor $\mathbf{X}$ (Fig. 1b). Furthermore, comboFM makes it possible to integrate any auxiliary data of the drugs and cell lines, such as chemical descriptors in the form of molecular fingerprints of drug compounds, gene expression profiles of the cancer cell lines and concentration values tested for the drugs.

For the learning algorithm, the data tensor $\mathbf{X}$ is flattened into a two-dimensional array (Fig. 1c), where each row vector $\mathbf{x}$ identifies a single entry in the original tensor. Given the associated responses $y_i$ in the training data, comboFM model is learned using factorization machines (FMs). Higher-order FMs learn a non-linear regression model from the input features ($\mathbf{x}$) to the output ($y$) by estimating a regression weight $w_{i_1,\ldots,i_t}$ for each combination of input features $x_{i_1} \cdot x_{i_2} \cdots x_{i_t}$, where $t$ is the order of the interaction. However, instead of estimating the weights $w_{i_1,\ldots,i_t}$ separately as in polynomial regression, FMs approximate the weights using factorized parametrization (Fig. 1d), where the weights are coupled through multiplication of latent factors learned by the FM. This approach avoids the computational and statistical problems that would result from directly estimating the weight tensor $\mathbf{W}$. In addition, the coupling of the weights allows effective learning in situations where the data tensor is sparsely populated.

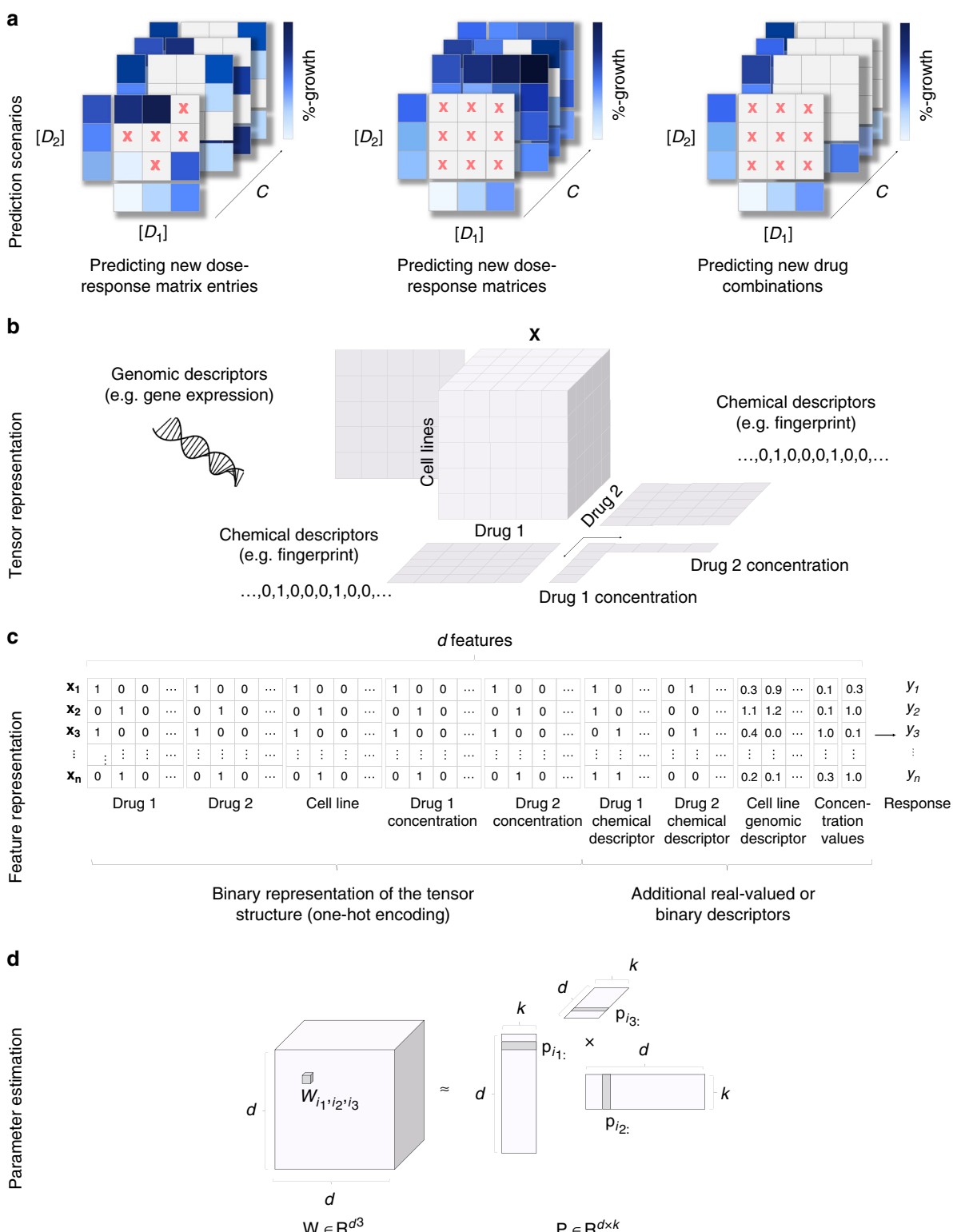

**Fig. 1 Overview of the comboFM framework for the prediction of drug-dose combination effects.** **a** Three prediction scenarios are considered: filling in missing entries in partially tested dose–response matrices, predicting a complete dose–response matrix in a new cell line, and making predictions for a completely new drug combination not tested so far in any cell line. **b** In each prediction scenario, the experimentally measured dose–response matrices are compiled into a fifth-order tensor **X** indexed by drugs ($D_1$, $D_2$), drug concentrations ($[D_1]$, $[D_2]$) and cell lines ($C$), and genomic and chemical descriptors are integrated into the prediction model. **c** The structure of the tensor underlying the drug combination dose–response matrix data is one-hot encoded into a single feature matrix together with the additional chemical and genomic descriptors. **d** The model parameters $w_{i_1, i_2, \ldots, i_t}$, for a $t$th order combination ($t = 3$ depicted) of features $i_1, \ldots, i_t$ are approximated using factorized parametrization $w_{i_1, i_2, \ldots, i_t} \approx \sum_{s=1}^{k} p_{1s} p_{2s} \ldots p_{ts}$ (see "Methods"). $d$ denotes the total number of features and $k$ is a hyperparameter defining the rank of the factorization.

**Accurate drug combination response predictions by comboFM.** To systematically evaluate the comboFM model, we used the anticancer drug combination response data from the NCI-ALMANAC study[4]. To enable various splits of data into different cross-validation folds as required by the different prediction scenarios and to keep the computational complexity manageable, we considered a subset of the data consisting of 50 unique FDA-approved drugs (Supplementary Table 3) in 617 distinct combinations screened in various concentration pairs across all the 60 cell lines originating from 9 tissue types[23]. In this data subset, a total of 333,180 drug combination response measurements and 222,120 monotherapy response measurements of single drugs are available in the form of percentage growth of the cell lines (see "Methods"). To computationally quantify the performance of comboFM in predicting drug combination responses and optimize the model parameters, we performed a $10 \times 5$ (10 outer folds, 5 inner folds) nested cross-validation (CV) procedure under the three prediction scenarios (see "Methods"). The order of the feature interactions modeled by the FM was set to $m = 5$, according to the order of the underlying tensor.

To investigate the benefit of considering higher-order feature interactions, we also performed experiments using both second order formulation of FMs and first order FMs (corresponding to ridge regression). To further benchmark the predictive performance of comboFM, we applied random forest (RF) as a reference model, a widely-used machine learning model that is based on a rather different learning principle, and has previously been used for modeling drug combination effects[24–28], including the winning method of the recent AstraZeneca-Sanger drug combination prediction DREAM Challenge[19]. The cross-validation folds were held fixed throughout the experiments to ensure a fair comparison. We assessed the predictive performance of the methods using root mean squared error (RMSE), as well as Pearson and Spearman correlation between original and predicted dose–response matrices.

By leveraging the multi-way interactions present in the underlying high-dimensional drug combination space across drugs, drug concentrations, and cancer cell lines, the 5th order comboFM demonstrated high predictive accuracy in all the three prediction scenarios (Fig. 2), outperforming the random forest reference ($p < 10^{-10}$ in all prediction scenarios, two-sided Wilcoxon paired signed rank sum test, $N = 666,360$). In the scenarios of predicting new dose–response matrix entries and new dose–response matrices, the 5th order comboFM obtained a Pearson correlation of 0.97, and even in the new drug combination prediction scenario, the 5th order comboFM obtained a Pearson correlation of 0.95. The 5th order comboFM was also markedly more accurate than both the 1st- and 2nd order comboFMs in all the three scenarios. Similar relative performance of the methods was also observed using Spearman correlation and RMSE (Fig. 2). In addition, the distribution of the predictions by 5th order comboFM followed that of the measured responses most accurately (Supplementary Fig. 1).

In addition to the global predictive performance of the methods, we analyzed also their performance in different tissue types and across the various types of drug combination therapies (Fig. 3 and Supplementary Figs. 2–4 and Supplementary Table 1). In all the three prediction scenarios (Fig. 3a–c), comboFM showed the highest average prediction accuracy in each of the tissue types, and also the smallest variance across the tissue types. The combination response in colon cancer appeared marginally more difficult to predict than the other tissue types, which is likely explained by higher variation in the colon cancer response data, as the number of colon cancer cell lines was similar to the other tissue types and thus the marginally inferior performance is

unlikely to stem from limited data quantity. Nevertheless, the 5th order comboFM was still the most accurate method also in colon cancer cell lines. Furthermore, comboFM was shown to provide high accuracies across various types of combination therapies (chemotherapies, targeted therapies, and other therapies, such as hormonal therapies) (Fig. 3d–f). The combination therapies involving drugs from the Other class include the smallest number of observations, explaining their reduced predictive accuracy with all the methods.

To further validate the performance of the 5th order comboFM, we also evaluated its predictive accuracy in the remaining part of the NCI-ALMANAC data that was not used in the cross-validation, consisting of 4737 distinct drug combinations. The model was trained on the full development dataset of 617 drug combinations as well as the monotherapy responses of the single drugs in the validation set, and the trained model was then used for predicting responses of the 4737 drug combinations in the validation set across the various cell lines. 5th order comboFM demonstrated high predictive accuracy also in this validation set (Supplementary Figs. 5 and 6), with Pearson correlation of 0.91 even for combinations where neither drug had previously been observed in any other combination, i.e. only the monotherapy responses of the individual drugs in the combination were available to the model.

**Synergy scores can be recovered with high accuracy based on the predicted dose–response matrices.** As the interest in drug combination experiments often lies in discovering the most synergistic drug combinations, we also quantified drug combination synergies based on the dose–response matrices predicted with comboFM. As a synergy quantification model, we applied NCI ComboScore (see "Methods")[4], computed over the complete predicted dose–response matrices. Although drug combinations with an NCI ComboScore above zero are technically defined to be synergistic, combinations with highly synergistic effects are typically considered as more attractive candidates for further experimental validation. Therefore, we labeled the extreme synergistic drug combinations (observed NCI ComboScore value in the top 10%) as the positive class and the remaining combinations, including lowly synergistic, additive, and antagonist combinations, as the negative class.

Drug combination synergy scores were recovered with a high accuracy from the dose–response matrices predicted by the 5th order comboFM in all three prediction scenarios, significantly outperforming the other compared methods (Supplementary Fig. 7). Importantly, the drug combination synergies could be accurately computed based on the predicted dose–response matrices using 5th order comboFM even in the challenging scenario of predicting new drug combinations, with a Pearson correlation of 0.72 ($p < 10^{-10}$, two-sided $t$-test, $N = 74,040$) between the observed and predicted NCI ComboScores. In the task of discriminating highly synergistic drug combinations, the 5th order comboFM obtained a high area under the receiver characteristic operator curve (AUC) of 0.91 in the new drug combination prediction task (Supplementary Fig. 8). The discrimination accuracies were at high level in each prediction scenario, and when using various top-% extreme synergy combinations (Supplementary Fig. 8).

**Experimental validation of the most synergistic predicted drug combinations.** To further demonstrate the ability of comboFM to predict novel and robust drug combinations, the model was trained using all the available dose–response measurements in the development dataset, and the trained comboFM was then used to predict dose–response matrices for remaining

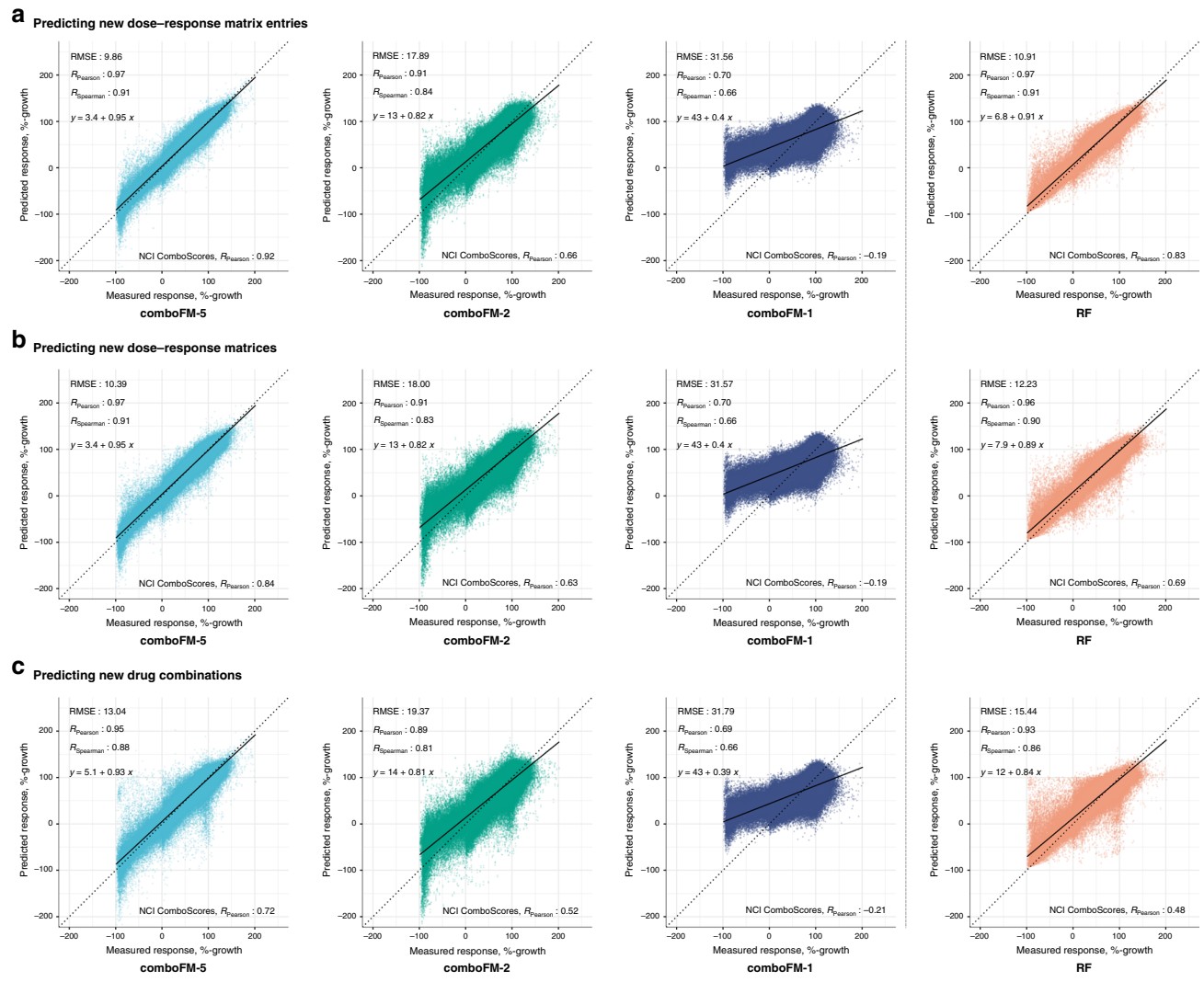

**Fig. 2 Predictive performance of 5th (comboFM-5), 2nd (comboFM-1) and 1st order comboFM (comboFM-1), and random forest (RF) as scatter plots between the measured and predicted dose–response matrices.** The responses were measured by percentage growth in the three prediction scenarios: **a** new dose–response matrix entries, **b** new dose–response matrices, and **c** new drug combinations. Root mean squared error (RMSE), Pearson correlation (RPearson) and Spearman correlation (RSpearman) for the drug combination response prediction are reported as averages over 10 outer CV folds. The Pearson correlation of the NCI ComboScores is reported as an average over all computed NCI ComboScores, computed based on the predicted dose–response matrices. Trend line and its equation are shown for each scatter plot.

unmeasured drug combinations across all the 60 cell lines, which resulted in a total of 10,320 predicted complete dose–response matrices. Experimental validation was performed subsequently on a subset of 16 drug combinations specific for 4 cell lines (Supplementary Table 2), where high synergy was predicted by comboFM. These combinations were selected to mainly involve molecularly targeted therapies, as recent interest has increasingly evolved toward targeted agents over the standard cytotoxic chemotherapies. In particular, we focused on cancer-specific drug combinations which were predicted to have highly synergistic effects only in a subset of all the cell lines and tissue types. This poses a more challenging task than identifying broadly toxic combinations that kill most cancer cells, but which may also induce severe toxicities in the healthy cells. As in the previous experiments, we considered as highly synergistic those combinations with an observed NCI ComboScore values in the top 10% in a particular tissue type.

The results of the experimental validation of 16 drug–drug–cell line triplets are summarised in Fig. 4, using the Bliss model to quantify the observed synergy. The background histogram shows

a distribution of an in-house drug combination dataset, consisting of 60 drug combinations tested against 16 KRAS-mutants pancreatic ductal adenocarcinoma cell lines. Since the combinations in the reference set were not randomly-selected, the background synergy distribution shows a slight positive bias; however, since the assay was the same as the one used for the experimental validation of comboFM predictions ("Methods"), it is expected to provide a valid reference distribution for statistical evaluations. All the drug combinations predicted by comboFM were validated as synergistic, when considering positive Bliss score as evidence for a degree of synergy ($p < 10^{-4}$, binomial test against the background distribution). Importantly, 9 out of 16 combinations had a Bliss synergy score higher than 90% of the background distribution ($p < 10^{-5}$, binomial test). In addition to Bliss synergy score, we also computed the synergy scores using three other popular synergy models: Loewe, highest single agent (HSA) and zero-interaction potency (ZIP) scores (Supplementary Figs. 9 and 10). These results demonstrate the robustness of the comboFM predictions across various experimental setups and synergy scoring models.

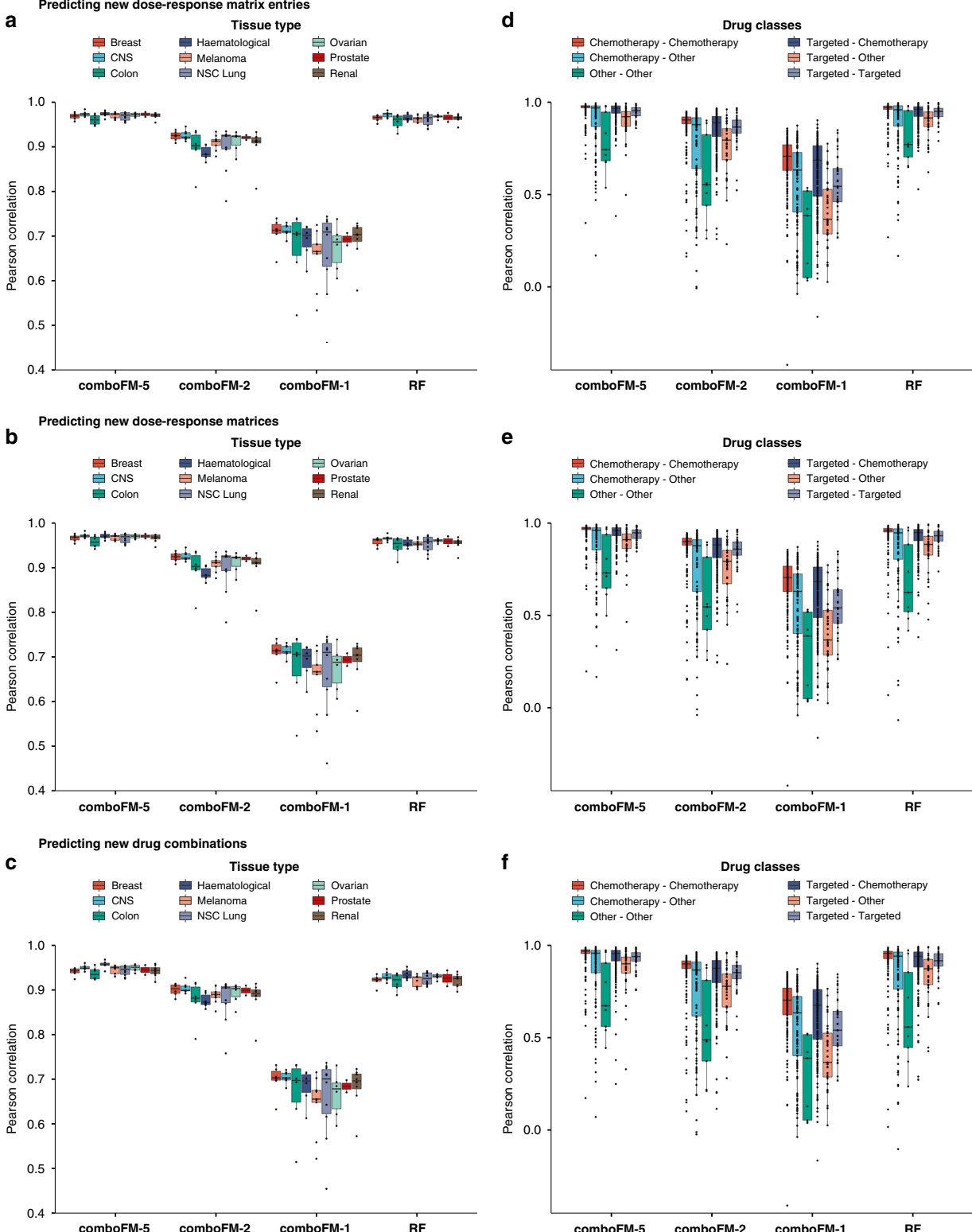

**Fig. 3 Predictive performance of 5th (comboFM-5), 2nd (comboFM-2) and 1st order comboFM (comboFM-1), and random forest (RF) across tissue types and drug classes in the three prediction scenarios. a–c** tissue types. **d–f** drug classes. The three prediction scenarios are depicted as follows: **a**, **d** predicting new dose–response matrix entries, **b**, **e** predicting new dose–response matrices, and **c**, **f** predicting new drug combinations. Further information on the drug classes can be found in Supplementary Table 3. In the boxplots, the horizontal lines drawn in the middle denote the median, and the lower and upper hinges correspond to the 25th and 75th percentiles, respectively. The upper and lower whiskers denote the largest and smallest values, respectively, no further than 1.5 times the inter-quartile range (IQR). The points that are not included between the whiskers are outlier predictions.

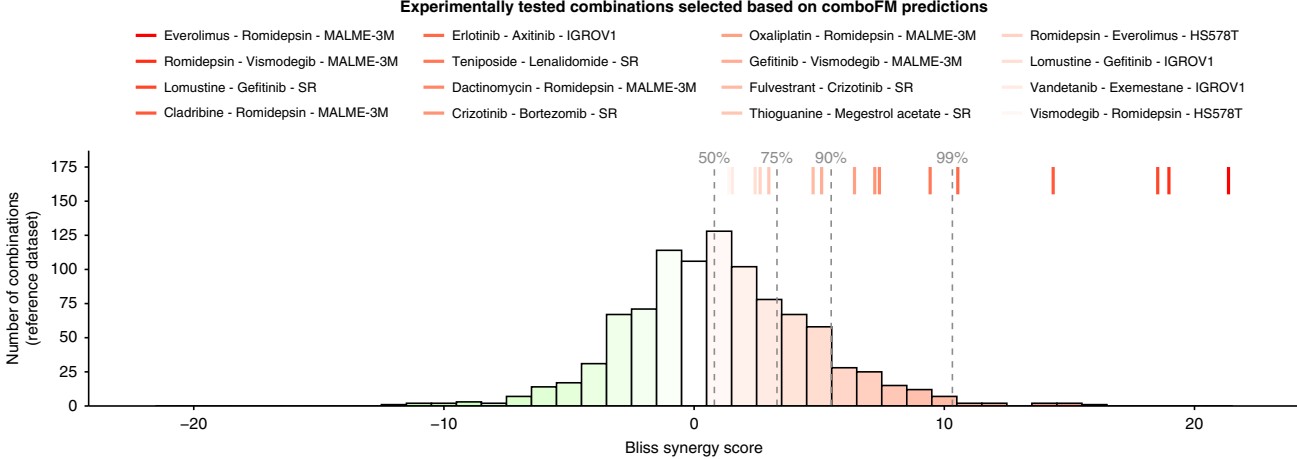

**Fig. 4 Measured drug combination synergy scores in the experimental validation.** In-house experimental validation of 16 selected predictions in specific cell lines are shown as colored lines (on top), and the histogram shows a background distribution from in-house reference dataset that comprises of 60 drug combinations tested against 16 KRAS-mutants pancreatic ductal adenocarcinoma cell lines (see "Methods"). The synergy was quantified using Bliss independence score for the most synergistic area of the dose–response matrix (see Supplementary Fig. 9 for other synergy scores). The color scale corresponds to the Bliss scores (green—antagonistic response, white—independent response, red—synergistic response). Dashed lines denote the percentiles of the background distribution obtained using the same experimental setup.

Among others, comboFM predicted a particularly high level of synergy for the combination between anaplastic lymphoma kinase (ALK) inhibitor crizotinib and proteasome inhibitor bortezomib in lymphoma cell line SR. In addition to our in-house experimental validations, this finding was further validated in external measurements in the NCI-ALMANAC data that were not used as part of comboFM training data. The ALK inhibitors are effective against cancers harboring ALK fusions. The SR cell line carries the NPM1-ALK fusion, which is the first ever discovered ALK fusion in large-cell lymphoma[29]. Bortezomib is approved for mantel cell lymphoma supporting its potential in lymphoma treatment. It is likely that two even mildly effective inhibitors when used in combination may enhance the inhibition effect and potentially overcome monotherapy resistance. Notably, comboFM made this prediction without knowledge of the ALK fusion status of the SR cell line, i.e., this biological rationale was not available for the model. The prediction of high synergy between the first-generation inhibitors of ALK and proteasome for lymphoma cell lines highlights the potential of comboFM to predict biologically plausible combination effects.

The comboFM model identified also another unique drug combination effective against the SR cell line, the combination of EGFR inhibitor gefitinib with an approved chemotherapy lomustine for lymphoma treatment. One of the mechanisms inducing resistance to ALK inhibitors is activation of EGFR, as they signal through similar downstream pathways. Brigatinib, a dual ALK/EGFR inhibitor, is therefore being explored in clinical settings against lymphoma and lung cancer patients (NCT01449461). Our comboFM method predicted combination partners to extensively explored ALK and EGFR inhibitors for lymphoma, which we were able to also validate in the experimental setting (Fig. 4). These examples show the potential of comboFM to identify novel combinations of both targeted and cytotoxic treatments, that individually are already used as lymphoma treatments, and therefore are likely to have acceptable toxicity profiles in clinical applications.

## Discussion

Given the enormous number of conceivable drug and dose combinations, computational approaches are needed to accelerate the experimental work by providing guidance toward identifying the most promising drug combinations for further experimental validation. While large datasets of drug combination dose–response matrices have already been tested in the lab, extensive gaps still remain in the combinatorial space among both targeted and non-targeted therapies, as well as hormonal and immunotherapies. Here, we have presented a novel machine learning framework, comboFM, for large-scale systematic prediction of drug combination effects in human cancer cell lines. The obtained results demonstrate that comboFM can leverage predictive higher-order relationships between drugs, drug concentrations, and cancer cell line responses, which were missed when using random forest and simpler approaches, including 1st and 2nd order formulation of comboFM. Importantly, comboFM can accurately generalize the predictions also for new drug combinations not observed in the training space, which enables one to systematically predict dose–response matrices also for so far untested drug combinations formed by the individual drugs in the training set. This will provide guidance on repositioning the drugs into new combinations. We also demonstrated that comboFM consistently obtains high prediction performance across various tissue types and classes of drug combination therapy. In addition, 5th order comboFM was 3 times faster to train compared to the random forest reference when run on the same CPU and considering relatively conservative amount of 200 training epochs for training the comboFM model (Supplementary Table 2). Further performance advantages were obtained by employing a GPU for training the 5th order comboFM model (34 times faster compared to random forest).

Modeling the drug combination effects first at the level of dose–response matrices and subsequently quantifying the level of overall drug combination synergy over the full matrix provides many benefits compared to approaches that directly aim at predicting the drug combination synergies. First of all, predicting the underlying dose–response matrices enables one to leverage all the information contained in the dose–response matrices and provides detailed information of the response landscape across various dose combinations. In addition, in the second stage, one is not limited only by a single synergy quantification model, but can explore the synergies using various models, hence gaining a more comprehensive view of the synergistic drug combination

landscapes[30]. Furthermore, understanding the drug combination effects both at the dose level as well as at the synergy level provides useful guidance for precision medicine efforts. For instance, combination synergies observed at lower doses are often better tolerated in the clinical practice. Furthermore, it has been shown that for most of the FDA-approved drug combinations, only little evidence of additivity or synergy was observed in pre-clinical models[31], highlighting that synergy is not always needed for clinical treatment success. However, it has also been argued that patient stratification based on predictive markers is likely to reduce variability in clinical therapy responses, and contribute to achieving truly synergistic responses to combination treatments[32].

In-house experimental validations of the top-synergistic combinations predicted using the NCI-ALMANAC data demonstrated that the comboFM predictions are robust also to the experimental setup. The in-house assay had many experimental differences when compared to the combination assay used to profile the NCI-ALMANAC development dataset. In particular, the in-house assay measured the drug combination responses in the form of percentage inhibition, instead of percentage growth that is used in the NCI-ALMANAC assay. Therefore, we could not calculate the NCI ComboScore for the experimental validations, but instead scored the combinations using four popular synergy models (Supplementary Figs. 9 and 10). As an example, comboFM predicted a pivotal role of histone deacetylase (HDAC) in melanoma cell line MALME-3M, thereby suggesting potential of HDAC inhibition against melanoma. In particular, various combinations with HDAC inhibitor romidepsin were predicted to be effective against BRAF-mutants melanoma cell line MALME-3M, which also held true in the experimental settings (Fig. 4). Even though most of the drugs in the romidepsin-combinations have already been explored in different combinations to target melanoma[33,34], the combinations predicted by comboFM have remained unexplored against melanoma, and warrant further investigation. Individually, each of these inhibitors have shown promising results in pre-clinical or clinical settings against melanoma, further supporting their use in combination therapies.

Even though the main objective of this work was to develop and carefully validate the comboFM model in cancer cell lines as an accurate methodology for systematic prediction of drug combination responses for biological discovery, we note that many of the drugs identified by comboFM have been or are currently being explored in clinical settings against the specific cancer type, either as single agents or in combination with other drugs (see Supplementary Table 5). For instance, HDAC inhibitor vorinostat is being tested against BRAF-mutant advanced melanoma in an ongoing clinical trial (ref. [35]; NCT02836548). Similarly, mTOR inhibitor everolimus is shown to selectively target BRAF-mutant melanoma in acidic condition[36]. In an ongoing clinical trial, mTOR inhibitors everolimus or temsirolimus in combination with BRAF inhibitor are being investigated against BRAF-mutant advanced solid tumors (NCT01596140). SMO-inhibitor vismodegib blocks Hedgehog pathway which regulates the skin growth. In case of medulloblastoma, HDAC inhibitors are active against even SMO-inhibitor resistant cell lines[37]. Hence, concurrent use of HDAC- and SMO- inhibitors holds a promising strategy to target melanoma, as predicted by romidepsin and vismodegib combination (Fig. 4). In the same line of rationale, combining HDAC inhibitor with DNA damaging agents, such as oxaliplatin, dactinomycin, and cladribine, holds strong promises and are explored in different pre-clinical and clinical settings[33,34,38,39].

These case examples already unveil the potential of our method for predicting combinations with translational potential, although these findings warrant further validation in proper clinical trials.

Furthermore, once the model accuracy has been confirmed in the cell line resources, we envision that the carefully validated model will be applicable also to data from individual cancer patients, thereby providing means for tailoring effective combinations in precision oncology applications. For selected cancer types, such as haematological malignancies, molecular and drug response profiling data are becoming available from patient-derived primary cells that can be used for training cancer type-specific prediction models[40,41]. Once similar data from other cancer types becomes available, comboFM will enable also pan-cancer analyses, similar to the current analyses in the NCI-ALMANAC cell lines. We found that many of the combinations predicted in the NCI-ALMANAC cell lines have actually already been tested in clinical trials (Supplementary Table 5). Interestingly, most of the combinations are tested in different indications than what was predicted based on the cell lines, suggesting further drug repurposing opportunities. The comboFM predictions require input data that start to be routinely available in many functional precision medicine studies, making it therefore broadly applicable for many cancer types and therapy classes.

In the present study, we assumed that one knows the monotherapy responses of single drugs prior to predicting the combination responses, as in practice it is often needed to know the concentration ranges and potencies of the single drugs (i.e., dose–response curves) in order to know which dose combinations should be used in combination testing, and also how potent the compounds are individually. comboFM strongly benefits from this information due to its capability to interpolate in the space of dose–response matrices through the computation of latent factors representing similarly behaving drug combinations from the response tensor alone (similarly to recommender systems grouping users by the movies they have liked in the past), while the drug and cell line descriptors merely fine-tune the predictions. It is plausible that by careful experimental design, one could minimize the number of monotherapy responses needed for accurate dose–response matrix prediction[42] whilst maintaining the accuracy of the comboFM model, which we leave as an interesting future research topic. However, in a scenario where one would like to perform predictions for completely new molecules with no prior monotherapy or combination response data in any cell line, the computed latent factors are no longer helpful, and none of the methods could perform well with the current design (Supplementary Fig. 13). This limitation of the methodology in such scenarios could potentially be addressed by more extensive feature engineering or by developing models that are specialized for the case of predicting dose–response matrices for combinations of completely new drugs.

As with any high-throughput pre-clinical data, the cell line drug response profiles may show inconsistency in experimental outputs across the same cell line-treatment pairs[43]. Therefore, we argue that it is important to develop and initially evaluate the prediction models in large enough and standardized cell line resources, such as NCI-ALMANAC, to avoid any reproducibility issues in the development phase. We further tested the model predictions using distinct experimental setups in the same cell lines to show that the predictions were robust enough against such biological and technical variability.

In conclusion, given the high cost of the experimental screening of drug combinations, comboFM has the potential to provide time- and cost-effective means toward prioritizing the most promising drug combinations for further pre-clinical or clinical studies. The accurate and robust drug combination response predictions provide a promising approach to streamline the development and expansion of combination therapeutics in personalized cancer treatment. This could ultimately accelerate

the clinical use of combination therapeutics to combat acquired drug resistance and to increase therapeutic efficacies.

## Methods

**Higher-order factorization machines**. comboFM uses higher-order factorization machines (HOFM)[20,21] for predicting the drug–drug combination responses. HOFMs are non-linear regression models learned with a training set of examples

$$\{(\mathbf{x}_1, y_1), (\mathbf{x}_2, y_2), ..., (\mathbf{x}_n, y_n)\}$$

of feature vectors $\mathbf{x} \in \mathbb{R}^d$ and output labels $y \in \mathbb{R}$.

A trained HOFM models the output $y \in \mathbb{R}$ as a function of single, pairwise, and higher-order interactions between input features up to order $m$:

$$\hat{y}(\mathbf{x}) := \sum_{i=1}^d w_i x_i + \sum_{1 \le i < i' \le d} w_{i,i'} x_i x_{i'} + ... + \sum_{1 \le i_1 < ... < i_m \le d} w_{i_1, i_2, i_m} \cdot x_{i_1} x_{i_2} ... x_{i_m}.$$

(1)

The first term corresponds to a linear model, and all parameters $w_i$ are independently estimated. The higher-order parameters are, on the other hand, estimated in a factorized form

$$w_{i,i'} = \langle \mathbf{p}_i^{(2)}, \mathbf{p}_{i'}^{(2)} \rangle$$

(2)

$$w_{i_1, i_2, i_t} = \langle \mathbf{p}_{i_1}^{(t)}, \mathbf{p}_{i_2}^{(t)}, ..., \mathbf{p}_{i_t}^{(t)} \rangle, t = 3, ..., m$$

(3)

where $\mathbf{p}_i^m \in \mathbb{R}^k$ denotes the $m$th order factor weight of feature $i$, $k$ is the hyperparameter defining the rank of the factorization, and

$$\langle \mathbf{a}_1, \mathbf{a}_2, ..., \mathbf{a}_m \rangle = \sum_{s=1}^k a_{1s} a_{2s} \cdots a_{ms}$$

(4)

denotes a generalized inner product of $m$ vectors $\mathbf{a}_i \in \mathbb{R}^k, i = 1, ..., m$ that generalizes the usual pairwise inner product $\langle \mathbf{a}, \mathbf{b} \rangle = \mathbf{a}^T \mathbf{b}$ to sets of $m$ vectors.

The factor weights are collected into matrices $\mathbf{P}^{(m)} = (\mathbf{p}_1^m, ..., \mathbf{p}_d^m)^T \in \mathbb{R}^{d \times k}$. The factorized parametrization drastically reduces the number of estimated parameters from $O(d^m)$ (all feature combinations have their own parameter) to $O(kdm)$ ($m-1$ factor matrices of dimension $d \times k$). In principle HOFMs allow an unique rank $k_t$ for each order $t = 2, ..., m$. In the above description and in our experiments, we used uniform rank $k = k_2 = ... = k_m$.

FMs are based on the assumption that the effect of pairwise and higher-order feature interactions has a low rank and allows FMs to estimate reliable parameters even under highly sparse data. Hence, the co-occurrence of $x_i$ and $x_{i'}$ does not need to be observed in order to learn $w_{i,i'}$: the factors $\mathbf{p}_{i'}$ and $\mathbf{p}_{i'}$ can be learned by interacting with other dimensions and the dot product of $\mathbf{p}_{i'}$ and $\mathbf{p}_{i'}$ still gives $w_{i,i'}$. This is extremely useful in the case of high-dimensional drug combination data where the input tensor is typically very sparse, and thus allows to make reliable inferences of the responses to new drug combinations whose individual components have still been observed in other combinations elsewhere in the training tensor. Compared to standard matrix factorization approaches, FMs provide additional flexibility by allowing integration of auxiliary data describing the drugs and cell lines, such as chemical and genomic descriptors.

The objective function of learning higher-order factorization machines is to minimize the regularized mean squared error

$$\min \frac{1}{n} \sum_{i=1}^n (y_i - \hat{y}_i(\mathbf{x}_i))^2 + \frac{\beta_1}{2} ||\mathbf{w}||^2 + \sum_{t=2}^m \frac{\beta_t}{2} ||\mathbf{P}^{(t)}||^2$$

(5)

where $\beta_1, ..., \beta_m > 0$ are regularization parameters. To limit the number of hyperparameter combinations to search, following the work by Blondel et al.[20], we set $\beta_1 = ... = \beta_m$, and a uniform rank $k = k_2 = ... = k_m$. In the experiments, we used a recent TensorFlow implementation of higher-order factorization machines[44].

On the NCI-ALMANAC data, increasing the order and rank of the factorization machine both improve the predictive performance (Pearson correlation) of the comboFM model (Supplementary Fig. 12). The predictive performance increases steeply until order 5, which matches the intrinsic order of the data tensor $\mathbf{X}$ (See Fig. 1b), and then continues to increase more slowly. The performance increase due to increasing rank of the factorization is rapid until around rank 50 and then continues to increase more slowly. There is no apparent overfitting even with factorization order as high as 10 and rank as high as 150.

**Synergy quantification**. As the interest often lies in discovering the most synergistic drug combinations, we quantify the drug combination synergies based on the predicted dose–response matrices. To compute the synergy scores, we apply the NCI ComboScore, which was introduced along with the NCI-ALMANAC dataset[4], originally modified from the Bliss independence score.

The NCI ComboScore for drug $A$ and drug $B$ is defined as the sum of the deviations between expected and observed responses over all concentrations $p$

and $q$:

$$y(A, B) = \sum_{p,q} (y_c(A_p, B_q) - y_e(A_p, B_q))$$

(6)

where $y_c(A_p, B_q)$ is the combination growth fraction of the cell line exposed to drug $A$ in concentration $p$ and drug $B$ in concentration $q$, and $y_e(A_p, B_q)$ is the expected growth fraction for the combination defined based on the monotherapy effects of drug $A$ and drug $B$ as follows:

$$y_e(A_p, B_q) = \begin{cases} \min(y_m(A_p), y_m(B_q)) \text{ if } y_m(A_p) \le 0 \text{ or } y_m(B_q) \le 0 \\ \frac{1}{150} (\tilde{y}_m(A_p) \cdot \tilde{y}_m(B_q)) \text{ otherwise} \end{cases}$$

(7)

where $y_m(A_p)$ and $y_m(B_q)$ denote the monotherapy effects of drug $A$ in concentration $p$ and drug $B$ in concentration $q$, respectively. We applied $\tilde{y}_m = \min(y_m, 150)$ that truncates the growth fraction at 150, with the threshold selected based on the histogram of the measured drug combination responses (Supplementary Fig. 11).

**Training setup**. In order to evaluate the predictive performance and optimize the model parameters under the three prediction scenarios, we performed a $10 \times 5$ (10 outer folds, 5 inner folds) nested cross-validation procedure. For all the factorization machine models, the rank parameter was optimized in the range $k = \{25, 50, 75, 100\}$ and the regularization parameter in the range $\beta = \{10^2, 10^3, 10^4, 10^5\}$. The order of the modeled feature interaction was set to 5 according to the order of the underlying tensor, as a compromise between the training time and prediction accuracy. The learning rate was set to 0.001 based on preliminary experiments and other parameters were kept in their default values. The number of trees of the random forest model was optimized in the range $\{32, 64, 128, 512\}$ and the fraction of features considered when looking for the best split (MaxFeatures) in the range $\{0.25, 0.5, 0.75, 1.0\}$.

As each input sample is represented by a single feature vector, in order to take the symmetry of the drug combinations into account, the samples were duplicated such that both of the drugs in a combination were included in both positions in the feature vectors. This informs the algorithm that the combination of drug A with drug B should be considered the same as the combination of drug B with drug A. The prediction accuracy of all the models was assessed using the same performance evaluation metrics: RMSE, Pearson correlation, and Spearman correlation.

**Evaluation of the prediction performance**. In this type of applications, the predictive performance is significantly affected by whether the training and test sets share the different components of the modeled interactions, and it is thus important to reliably quantify the prediction accuracy under practical application scenarios. Therefore, we evaluated the predictive performance of comboFM under three prediction scenarios: (a) new dose–response matrix entry prediction, (b) new dose–response matrix prediction and (c) new drug combination prediction (c.f. Fig. 1). For each scenario, we used dedicated nested cross-validation setups to ensure unbiased evaluation. In scenario (a), the predictions were made for individual held-out entries in dose–response matrices. The held-out entries were selected at random for each cross-validation fold. In scenario (b), the predictions were made for completely held out (dose–response matrix, cell line) pair, such that the same drug combination had still been measured in other cell lines. This scenario corresponds to a widely-used strategy in other computational works concerning drug combination synergy prediction, in which the predictions are made for new drug–drug-cell line triplets. In scenario (c), most challenging scenario of new drug combination prediction, the predictions are made for novel drug combinations outside the training space with no available combination measurements. In all prediction scenarios, we assumed that the monotherapy responses of the single drugs in the combination are known.

To computationally evaluate the prediction performance and optimize the model parameters, we performed a nested cross-validation procedure. In the first prediction scenario of new dose–response matrix entry prediction, the cross-validation folds were formed by simply random sampling from the tensor entries. In the second prediction scenario concerning new dose–response matrices, the folds were created by randomly sampling on the level of dose–response matrices, i.e., if a drug pair-cell line triplet $(x_{d_1}, x_{d_2}, x_c)$ belonged to the test set, the training tensor did not include any entry involving the triplet $(x_{d_1}, x_{d_2}, x_c)$. In the third scenario of new drug combination prediction, the random sampling was performed on the level of drug pairs and all the entries involving the test drug pairs were held out from the training set, i.e., if a drug pair $(x_{d_1}, x_{d_2})$ belonged to the test set, the training tensor did not contain any entry involving the pair $(x_{d_1}, x_{d_2})$. Furthermore, we ensured that the individual drugs in the left out drug pairs are still observed individually in other combinations in the training set, which enables the model to learn from the way the individual drugs in the held out combinations act in other combinations.

**Drug combination anticancer activity dataset**. The drug combination anticancer activity dataset was obtained from a recent NCI-ALMANAC study[4], which is the largest available drug combination dataset to date. The original dataset covers over 5000 combinations of roughly 100 small molecule drugs screened against 60 cell

lines in various concentrations, containing over 3 million response measurements. The drugs included in the dataset are FDA-approved oncology drugs with proven activity and established safety profiles. The cell lines represent human tumor cell lines from the NCI-60 panel, originating from 9 different tissue types.

To reduce the computational complexity, we selected a subset of the NCI-ALMANAC dataset by randomly sampling 50 drugs (Supplementary Table 3) from the original set of drugs, ensuring that the distribution of the subset of drug combination responses matched to that of the original one. Furthermore, we selected drug combinations for which complete measurements across all the 60 cell lines were available. As a result, we obtained a dataset for our experiments consisting of 617 drug combinations of 50 unique drugs, screened in 45 unique concentrations against 60 cell lines, containing 333,180 response measurements for combinations and 222,120 measurements for monotherapies, measured by percentage growth of the cell line with respect to a control. Each drug combination in the dataset had been screened using $4 \times 4$ dose–response matrix design.

**Data representation**. Defining an informative input feature representation of the underlying data is essential to take the full advantage of comboFM and FMs in general. By defining appropriate input features, FMs have been shown to have the representation power encompassing a variety of matrix and tensor factorization models from standard models to more specialized ones[21,22]. Hence, by learning FMs, all the subsumed factorization models can also be learned.

In order to represent the structure of the tensor underlying the drug combination response data as single input feature vectors, one-hot encoding is used. Here, the input feature vectors $\mathbf{x}$ are divided into five different groups corresponding to the different modes of the tensor: two sets of drugs, their concentrations, and a cell line. In each group, exactly one value is set to 1 and the rest to 0, with 1 denoting the instance that is present in the corresponding interaction:

$$\mathbf{x} = \left( \underbrace{0, ..., 0, 1, 0, ..., 0}_{|\text{Drugs}|} \underbrace{0, ..., 0, 1, 0, ..., 0}_{|\text{Drugs}|} \underbrace{0, ..., 0, 1, 0, ..., 0}_{|\text{Concentrations}|} \underbrace{0, ..., 0, 1, 0, ..., 0}_{|\text{Concentrations}|} \underbrace{0, ..., 0, 1, 0, ..., 0}_{|\text{Cell lines}|} \right). \quad (8)$$

As the feature vector is non-zero only for the pair of drugs, drug concentrations, and cell line present in the corresponding interaction, all the other interactions in the FM model vanish and the model corresponds to standard factorization models involving categorical variables. However, whereas standard factorization models are limited to categorical input data only, comboFM and FMs can also incorporate auxiliary features in addition to the information of the interacting elements, which can further aid the prediction task, particularly when making predictions outside the training space. In this work, we used chemical descriptors of molecules and genomic descriptors of cell lines (see below for details).

**Chemical descriptors**. As chemical descriptors, we integrated molecular fingerprints, binary vectors which are designed to represent the structure of a molecule as a series of bits, each one representing the presence or absence of a particular substructure. We selected a popular fingerprint of type 'estate', consisting of 79 bits corresponding to the E-State atom types originally defined by[45], obtained from the rcdk R package[46]. Fingerprint bits with zero variance across the dataset were further removed, resulting in remaining 34 bits for the two sets of drugs.

**Genomic descriptors**. As genomic descriptors, we incorporated gene expression profiles of the cancer cell lines, obtained from the rcellminer R package[47]. The gene expression profiles were measured with five different platforms (four Affymetrix arrays and an Agilent Whole Human Genome Oligo array) and a combined average z-score was reported as a combined gene expression for a gene. To reduce the dimensionality of the resulting feature matrix, we selected 0.5% of the genes with the highest variance across the samples, resulting in 78 gene expression values for each cell line.

**Cell lines**. Early passage cells lines purchased from ATCC (HS-578T & Malme-3M) and NCI-Frederick DCTD tumor/cell lines repository (SR & IGR-OV1) were used for drug combination screening. The cell lines were maintained at 37 °C with 5% $CO_2$ in a humidified incubator in their respective medium (see Supplementary Table 4a). All the reagents were purchased from ThermoFisher Scientific. All the cell lines were tested negative for mycoplasma. The test was based on the method described by Choppa et al.[48] and was performed as a service by the sample management laboratory of THL Biobank, Helsinki, Finland.

**Drug combination screening**. The drug combination testing experimental design was adopted from Gautam et al.[49]. Seven different concentrations in log3-fold dilution of two drugs were combined with each other in $8 \times 8$ matrix formats. Please refer to Supplementary Tables 4b and c for the dug information and combinations design, respectively. The compounds were plated to black clear bottom 384-well plates (Corning #3764) using an Echo 550 Liquid Handler (Labcyte). 100 μM benzethonium chloride (BzCl2) and 0.1% dimethyl sulfoxide (DMSO) were used as positive and negative controls, respectively. All subsequent liquid handling was performed using MultiFlo FX multi-mode dispenser (BioTek). The pre-dispensed compounds were dissolved in 5 μl of culture media and left in a plate shaker at room temperature for 30 min. Twenty microliter cell suspension (please refer to Supplementary Table 4a for cell line specific seeding densities) was dispensed in the drugged plates. After 72 h

incubation, 25 μl per well of CellTiter-Glo (Promega) reagent was added, and after 10 min of incubation at room temperature, luminescence (cell viability) was measured using PheraStar plate reader (BMG Labtech).

**Reporting summary**. Further information on research design is available in the Nature Research Reporting Summary linked to this article.

## Data availability

The NCI-ALMANAC dataset is publicly available from National Cancer Institute (NCI) at https://wiki.nci.nih.gov/display/NCIDTPdata/NCI-ALMANAC. The preprocessed data used in the computational experiments and in-house drug combination testing data for validating comboFM predictions are available at https://doi.org/10.5281/zenodo.4135059. Source data underlying the figures and display items are provided at https://doi.org/10.5281/zenodo.4135059 subdirectory source_data.

## Code availability

The code is available at https://doi.org/10.5281/zenodo.4129688.

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

## Acknowledgements

This work was supported by the Academy of Finland [ICT2023 programme grants 313268 to J.R.; 313266 to T.P. and 313267 to T.A. and grants 292611, 310507, 326238 to T.A.], the Cancer Society of Finland [T.A.]), the Sigrid Jusélius Foundation [T.A.], and Orion Research Foundation sr [P.G.]. The authors thank the FIMM HTB unit and especially Laura Turunen for their great help with the drug combination assays and Aleksandr Ianevski for his great help with the synergy scoring and the background distribution data for Fig. 4. The authors also acknowledge the computational resources provided by the Aalto Science-IT project as well as CSC - IT Center for Science, Finland.

## Author contributions

H.J., T.A., A.C., S.S., T.P., and J.R. designed the research. H.J., A.C., T.P., J.R., and S.S. developed computational methods and evaluation protocols. A.C., J.D. and H.J. performed computational evaluations. P.G. designed and performed experimental evaluation. H.J., A.C., P.G., J.R., and T.A. wrote the paper, contributed by T.P. and S.S.

## Competing interests

The authors declare no competing interests.
