## [Peer Review File · Nature Communications]

Reviewers' Comments:

Reviewer #1:

Remarks to the Author:

Julkunen et al. present comboFM, a novel machine learning framework for systematic modeling of drug-dose combination effects in a cell context-specific manner. It is based on the observation that the drug combination dose-response data can be compiled into a 5th-order tensor indexed by drugs, drug concentrations and cell lines. Thus, comboFM model is learned using high-order Factorization Machines (FMs).

comboFM is evaluated on three prediction scenarios: a) filling-in missing entries in partially-tested dose-response matrices, b) predicting a complete dose-response matrix in a new cell line, and c) making predictions for a completely new drug combination not tested. A randomly-generated subset of NCI-ALMANAC data is employed for the evaluation. This subset has a total of 617 drug combinations including 50 drugs at the various concentration pairs across all the 60 cell lines, thus comprising of 333,180 drug combination response measurements and 222,120 monotherapy response measurements of single drugs are available in the form of percentage growth of the cell lines. Random forest (RF) is used as a reference model. Nested cross-validation is properly carried out for each scenario and model.

The comboFM model was subsequently trained using all the data in the subset and then used to predict dose-response matrices for 172 unmeasured drug combinations across all the 60 cell lines, which resulted in a total of 10 320 predicted complete dose-response matrices. Experimental validation was finally performed on 16 of these 172 drug combinations specific for 4 cell lines, where high synergy score was predicted by comboFM using the Bliss metric. Weakly synergistic combinations (measured Bliss synergy scores between 1 and 21) were discovered in these cell lines. All the drug combinations predicted by comboFM were validated as synergistic, when considering the significance of the measured score against the distribution of scores from an in-house screen on pancreatic cancer cell lines.

Many issues, some of them critical, need to be addressed:

- 1) The selected random data subset only contains 617 of the 5000 drug combinations in NCI-ALMANAC, but they do not say why not all the data was used. Was it because the comboFM model is computationally expensive to train?
- 2) In any case, the time required to train the comboFM models of various orders and the RF model along with the employed computer should be presented and discussed.
- 3) The comboFM has a valid, yet important limitation. Scenario c) predict the responses of new combinations of the drugs, but the responses of those individual drugs are known to the model, which presumably provides an important advantage. In other words, if you applied comboFM to predict the synergies of all combinations of 10,000 drugs, you'll need to have the in vitro response of those 10,000 drugs on the same cell line in order to be able to make those predictions. This must be discussed in the paper. A 4th scenario should be investigated too, where neither the combination responses nor the responses of the two drugs forming the combination are in the training set.
- 4) In Figure 2, most model predictions overestimate negative responses: why?
- 5) NCI-Almanac measures synergies with comboscores, a metric that uses a modified Bliss model. This paper seems to use the unmodified Bliss model. To be able to compare to NCI-Almanac synergies, it should show comboscores too, exactly how they are defined by NCI-Almanac. Furthermore, predicted responses can be used to calculate the corresponding comboscores, so predicted vs original comboscores should be also shown in Figure 2.
- 6) While several hyperparameters are tuned for comboFM, only the number of trees is tuned for RF. More a fairer comparison, the maxfeatures hyperparameter of the RF python implementation should be tuned as well: any improvement?
- 7) In Figure 3, show any significant difference with respect to the corresponding comboFM-5 boxplot (i.e. same colors across models, so 3 p-values per color arising from the 3 pairwise model

comparisons). Some differences seem to be non-significant.

8) Importantly, the 5000-617 drug combinations in NCI-ALMANAC that were not used for training were not used either as a test set. No valid reason for this, as operating a trained model is much faster than training it. Exactly the same model that was used prospectively, i.e. that trained on all subset data, should be used to predict this large test set and that a plot with the predicted vs observed responses and another for the predicted vs observed comboscores be shown. By comparing these new plots with those in Figure 2, we will find out how predictive comboFM models really are. All these new codes and processed files should be also released for reproducibility.

9) Figure 4 shows Bliss synergy score. This should be NCI-ALMANAC's comboscore instead. Comboscores per cell line sometimes exceed 200 in synergy score, whereas here only mildly synergistic combinations are discovered. Given the claim that comboFM performs so well, it is odd that a plot with the predicted vs observed responses and another for the predicted vs observed comboscores from this prospective experiment are not presented. Please add and discuss these plots.

10) the 16 combinations validated prospectively were compared to an in-house, non-published, distribution of synergy scores to calculate p-values. I also find this odd given that the entire NCI-Almanac provides a much larger and relevant background distribution to calculate these p-values. This distribution should be used instead, always employing comboscores to permit the comparison.

Reviewer #2:

Remarks to the Author:

Julkunen, et al. present a method ComboFM that utilizes a higher order factorization machine on multidimensional tensor to predict drug combination effects in 3 different scenarios. The 3 scenarios are practical and they are common issues that researchers face. A nice feature of the FM is that information such as chemical descriptors and genomic descriptors can be integrated into the model. Overall, I am very supportive of this work. The paper is well written. In fact the only textual error I was able to find was a extra space before the period on page 13, "cross-validation fold ." The tasks that the authors propose have broad appeal to readers as these are very practical applications. The evaluation of the method and comparison to RF is done well and robustly using different datasets. The application of the FM is interesting and novel within a pretty well-studied research area. The code and the data are made publicly available. I very rarely ever accept a manuscript on first review, but I feel the work is publishable in its current form and I have not further suggestions that would improve or change the overall story that the authors present. The authors should be commended on their very strong paper on first submission.

Reviewer #3:

Remarks to the Author:

The manuscript 'comboFM: leveraging multi-way interactions for systematic prediction of drug combination effects' discusses the application a novel Machine Learning method to the task of preclinical response prediction. The manuscript has been very well-written with an easy-to-read flow and clear description of the results. The figures are largely self-explanatory and the code base on git is easy to understand as well.

While I agree with the premise, preclinical response modelling has been done extensively. It is the translatability to patients that is the major bottleneck and need of the hour. The authors should not only highlight this fact, but also discuss at length the translatability credentials of comboFM. This is currently missing. My major concern with the study is that it is not novel enough in terms of the scientific gain to the community. While the results are encouraging and the brief scientific discussion interesting, it is presented as another methods paper that surely shows statistical improvement in performance, but the deep biological insights that are needed for this very explored field and idea is missing. FMs are gaining prominence, as rightly pointed out by the

authors. But the biological/translational impact that is a significant improvement from previous published methods is needed to be more convincing of the validity of the presented predictive model.

My recommendation would be a major rewrite and resubmission highlighting the 'significant jump' in translational impact of this work over just being a synergy predictive model. All minor and major comments and feedback, along with key references, have been added as highlights and comments to the submitted PDF (see attached).

Reviewer #1 (Remarks to the Author):

Julkunen et al. present comboFM, a novel machine learning framework for systematic modeling of drug-dose combination effects in a cell context-specific manner. It is based on the observation that the drug combination dose-response data can be compiled into a 5th-order tensor indexed by drugs, drug concentrations and cell lines. Thus, comboFM model is learned using high-order Factorization Machines (FMs).

comboFM is evaluated on three prediction scenarios: a) filling-in missing entries in partially-tested dose-response matrices, b) predicting a complete dose-response matrix in a new cell line, and c) making predictions for a completely new drug combination not tested. A randomly-generated subset of NCI-ALMANAC data is employed for the evaluation. This subset has a total of 617 drug combinations including 50 drugs at the various concentration pairs across all the 60 cell lines, thus comprising of 333,180 drug combination response measurements and 222,120 monotherapy response measurements of single drugs are available in the form of percentage growth of the cell lines. Random forest (RF) is used as a reference model. Nested cross-validation is properly carried out for each scenario and model.

The comboFM model was subsequently trained using all the data in the subset and then used to predict dose-response matrices for 172 unmeasured drug combinations across all the 60 cell lines, which resulted in a total of 10 320 predicted complete dose-response matrices. Experimental validation was finally performed on 16 of these 172 drug combinations specific for 4 cell lines, where high synergy score was predicted by comboFM using the Bliss metric. Weakly synergistic combinations (measured Bliss synergy scores between 1 and 21) were discovered in these cell lines. All the drug combinations predicted by comboFM were validated as synergistic, when considering the significance of the measured score against the distribution of scores from an in-house screen on pancreatic cancer cell lines.

Many issues, some of them critical, need to be addressed:

1) The selected random data subset only contains 617 of the 5000 drug combinations in NCI-ALMANAC, but they do not say why not all the data was used. Was it because the comboFM model is computationally expensive to train?

Our response: We thank the reviewer for pointing this out. The reason was two-fold: first, we wanted to construct a development dataset where all the drug combinations were measured against all the 60 cell lines. This was deemed necessary to make the management of the various splits of data into different cross-validation folds more feasible as required by the different prediction scenarios. The second reason was indeed to keep the required computational costs manageable. We have now clarified this in the main text (page 5).

2) In any case, the time required to train the comboFM models of various orders and the RF model along with the employed computer should be presented and discussed.

Our response: We thank the Reviewer for the suggestion. We have now added a new Supplementary Table 2 that depicts the time consumption of comboFM models of various orders, in comparison to RF. This time complexity comparison shows that 5th order comboFM is 3 times faster to train compared to random forest when run on the same CPU and considering a relatively conservative amount of 200 training epochs for training comboFM. Further performance advantages can be obtained by employing a GPU for training the comboFM model (34 times faster compared to random forest). This result is described on page 10.

3) The comboFM has a valid, yet important limitation. Scenario c) predict the responses of new combinations of the drugs, but the responses of those individual drugs are known to the model, which presumably provides an important advantage. In other words, if you applied comboFM to predict the synergies of all combinations of 10,000 drugs, you'll need to have the in vitro response of those 10,000 drugs on the same cell line in order to be able to make those predictions. This must be discussed in the paper. A 4th scenario should be investigated too, where neither the combination responses nor the responses of the two drugs forming the combination are in the training set.

Our response: The rationale for assuming that the responses for the individual drugs are available was that it was considered reasonable to assume the practical requirement that one needs to know the concentration ranges and potencies of the single-drugs (i.e. dose-response curves) in order to know which dose combinations should be used in combination testing, and also how potent the compounds are individually. Therefore, the key insight behind comboFM is that by modeling the multiway interactions behind the dose-response data, comboFM effectively learns from the dose-response curves of individual drugs and the way the drugs in a combination act in other combinations and across various cell lines, without the need for extensive feature engineering. However, to address this comment, we have now added new supplementary Figure 10 for an experiment with 4th scenario, where neither the combination responses nor the responses of the two individual drugs forming the combination have been measured in any cell line in the training set, i.e. both the monotherapy responses of completely new molecules as well as their combination effects are being predicted. As expected, the results show that none of the methods perform well in this scenario. We have also added discussion on this limitation on page 12.

4) In Figure 2, most model predictions overestimate negative responses: why?

Our response: This is most likely due to the dataset being biased towards positive responses (please see new Supplementary Figure 1), which makes the models to focus on minimizing the prediction error in the positive end of the combination response spectrum. However, this effect is negligible in comboFM-5 which was also overall the most accurate model. We have now added new Supplementary Figure 1a showing the distributions of model predictions w.r.t the

measured responses. This comparison shows that 5th order comboFM follows the reference distribution most accurately (described on page 6).

5) NCI-Almanac measures synergies with comboscores, a metric that uses a modified Bliss model. This paper seems to use the unmodified Bliss model. To be able to compare to NCI-Almanac synergies, it should show comboscores too, exactly how they are defined by NCI-Almanac. Furthermore, predicted responses can be used to calculate the corresponding comboscores, so predicted vs original comboscores should be also shown in Figure 2.

Our response: We thank the reviewer for pointing out that our comparison of the prediction performance in terms of the ComboScores in the NCI-ALMANAC dataset was not clearly described in the original text. We have now added the correlation between the original and predicted ComboScores as an additional performance evaluation metric in Figure 2. In addition, we have added a new Supplementary Figure 3 that along with the original Supplementary Figure 4 further shows the performance of the methods in terms of original vs. predicted ComboScores in the different prediction scenarios, and clarified this comparison in the main text (page 6).

6) While several hyperparameters are tuned for comboFM, only the number of trees is tuned for RF. More a fairer comparison, the maxfeatures hyperparameter of the RF python implementation should be tuned as well: any improvement?

Our response: We thank the reviewer for this good suggestion. We have now properly tuned the RF model w.r.t. number of trees and MaxFeatures, and show its improved results in the revised Figures 2 and 3.

7) In Figure 3, show any significant difference with respect to the corresponding comboFM-5 boxplot (i.e. same colors across models, so 3 p-values per color arising from the 3 pairwise model comparisons). Some differences seem to be non-significant.

Our response: We thank the reviewer for the suggestion. We have now added a new Supplementary Table 1 listing the pairwise comparisons (and p-values) between the predictions of 5th-order comboFM and all the other models in each tissue type and drug class.

8) Importantly, the 5000-617 drug combinations in NCI-ALMANAC that were not used for training were not used either as a test set. No valid reason for this, as operating a trained model is much faster than training it. Exactly the same model that was used prospectively, i.e. that trained on all subset data, should be used to predict this large test set and that a plot with the predicted vs observed responses and another for the predicted vs observed comboscores be shown. By comparing these new plots with those in Figure 2, we will find out how predictive comboFM models really are. All these new codes and processed files should be also released for reproducibility.

Our response: We thank the reviewer for this good suggestion. We have now added a new Supplementary Figure 3 that depicts comboFM performance on the portion of data that was not used in the model development, where the model was trained using the full development set and the monotherapy responses of single drugs in the validation set. This result shows that 5th order comboFM can accurately predict the combination responses also in this validation set (Pearson correlation 0.908-0.924 between the measured and predicted responses). The new codes and processed data files have also been made publicly available at <https://github.com/aalto-ics-kepaco/comboFM> (codes) and <https://zenodo.org/record/3782333#.Xzf92y9h3yw> (data).

9) *Figure 4 shows Bliss synergy score. This should be NCI-ALMANAC's comboscore instead. Comboscores per cell line sometimes exceed 200 in synergy score, whereas here only mildly synergistic combinations are discovered. Given the claim that comboFM performs so well, it is odd that a plot with the predicted vs observed responses and another for the predicted vs observed comboscores from this prospective experiment are not presented. Please add and discuss these plots.*

Our response: The 16 selected combinations from comboFM predictions shown in Figure 4 were experimentally tested in-house using a different assay than that in NCI-ALMANAC, with the aim to assess how accurately the comboFM predictions can be validated also using different laboratory setup, compared to the NCI-ALMANAC training data, to prove its robustness to the experimental factors. In particular, in the in-house FIMM assay, the drug combination responses were measured in the form of percentage inhibition instead of percentage growth that is used in the NCI-ALMANAC assay. Hence, as the calculation of NCI ComboScore would require the combination responses to be measured in the form of percentage growth of the cell lines, computing NCI ComboScores for this experimental validation set was not possible. Therefore, we evaluated the combination synergies using four different and commonly used synergy scoring models (Bliss independence, Loewe additivity, zero interaction potency (ZIP) and the highest-single agent (HSA) models, see Supplementary Figures 6-7), which demonstrated the robustness of the comboFM predictions across various synergy models and experimental assays. We only included the Bliss independence score in the main Fig. 2, due to its popularity in drug combination studies and connection to the NCI ComboScore (and limited space in the main paper), but results with the other synergy models agree with those obtained with the Bliss model and provide further evidence of the synergy of these experimentally validated combinations. We feel this experimental validation is important for showing the robust performance of the comboFM, in addition to the computational validations in the NCI-ALMANAC data, and therefore we would like to keep these results even if NCI ComboScore could not be used.

10) *the 16 combinations validated prospectively were compared to an in-house, non-published, distribution of synergy scores to calculate p-values. I also find this odd given that the entire NCI-Almanac provides a much larger and relevant background distribution to calculate these*

p-values. This distribution should be used instead, always employing comboscores to permit the comparison.

Our response: As explained above, the 16 combinations were validated using a different assay compared to NCI-ALMANAC (see detailed comparison of the two assays below). Therefore, in order to provide a relevant reference distribution for the statistical evaluations, we used the additional in-house data screened with the same assay to make the combination synergies comparable between the comboFM validations and the reference distribution.

Both the in-house and NCI-ALMANAC drug combination assays were carried out in 384-well plates and CellTiter-Glo reagent was used to measure the cell viability. However, there were several differences between two assays; for instance, NCI-ALMANAC was carried out in 3X3 combination matrices, whereas our in-house pancreatic cancer screening (reference data) as well as experimental validations of the comboFM-predicted combinations were carried out in 8X8 settings to obtain a higher resolution. ALMANAC screens were carried out as 2 days end point assay whereas the in-house screens were for 3-days, since the optimal effect of most of the drugs are seen in 72h. The ALMANAC assay also lacked positive (cell killing) control and drug effects were quantified as growth percentage (with inclusion of time zero measurement), whereas in the in-house assay we included both positive and negative control, and the drug effects were therefore quantified as percent inhibition, normalized to both controls.

Reviewer #2 (Remarks to the Author):

Julkunen, et al. present a method ComboFM that utilizes a higher order factorization machine on multidimensional tensor to predict drug combination effects in 3 different scenarios. The 3 scenarios are practical and they are common issues that researchers face. A nice feature of the FM is that information such as chemical descriptors and genomic descriptors can be integrated into the model. Overall, I am very supportive of this work. The paper is well written. In fact the only textual error I was able to find was a extra space before the period on page 13, "cross-validation fold ." The tasks that the authors propose have broad appeal to readers as these are very practical applications. The evaluation of the method and comparison to RF is done well and robustly using different datasets. The application of the FM is interesting and novel within a pretty well-studied research area. The code and the data are made publicly available. I very rarely ever accept a manuscript on first review, but I feel the work is publishable in its current form and I have not further suggestions that would improve or change the overall story that the authors present. The authors should be commended on their very strong paper on first submission.

Our response: We thank the Referee for the encouraging comments.

Reviewer #3 (Remarks to the Author):

The manuscript 'comboFM: leveraging multi-way interactions for systematic prediction of drug combination effects' discusses the application a novel Machine Learning method to the task of preclinical response prediction. The manuscript has been very well-written with an easy-to-read flow and clear description of the results. The figures are largely self-explanatory and the code base on git is easy to understand as well.

While I agree with the premise, preclinical response modelling has been done extensively. It is the translatability to patients that is the major bottleneck and need of the hour. The authors should not only highlight this fact, but also discuss at length the translatability credentials of comboFM. This is currently missing. My major concern with the study is that it is not novel enough in terms of the scientific gain to the community. While the results are encouraging and the brief scientific discussion interesting, it is presented as another methods paper that surely shows statistical improvement in performance, but the deep biological insights that are needed for this very explored field and idea is missing. FMs are gaining prominence, as rightly pointed out by the authors. But the biological/translational impact that is a significant improvement from previous published methods is needed to be more convincing of the validity of the presented predictive model.

Our response: As was correctly pointed out by the Reviewer, comboFM provides significant improvements over the previous drug combination prediction models, as was demonstrated in various experimental setups in the manuscript. We would like to note that many of the existing prediction models cannot even deal with the more practical setups considered here, including making predictions for a completely new drug combination not tested so far in any cell line, whereas comboFM showed excellent prediction performance also in these more challenging setups that are more realistic from the practical point of view. A selected set of 16 combinations that were identified using comboFM were also experimentally validated in our subsequent laboratory experiments in selected cell lines, further demonstrating the biological relevance and robustness of the predictions. In the revised version, we now highlight several combinations that have clear biological rationale as well as combinations undergoing clinical trials (pages 11-12, and please see our responses below to the specific comments).

The main objective and impact of this work was to develop and carefully validate the comboFM in cancer cell lines as an accurate method by which one can make novel combinatorial discoveries faster, compared to the current, mostly hypothesis-based selection of drug combinations for further development. Even though finding biological rationale for the predicted and experimentally-validated combinations provides interesting biological insights, such biology-based prediction of combinations is a relatively lengthy and objective process, and has the risk of missing combinations supported by so-far unknown biology. This is why we believe data-driven prediction models, such as comboFM, provide more unbiased predictions of the most potent combinations for further testing. We indeed demonstrated how comboFM effectively predicts cell line-specific drug combinations which were also validated in experimental setting. Our systematic evaluations in multiple cancer cell lines and using a broad spectrum of targeted

and chemotoxic drugs clearly demonstrated the improved accuracy of comboFM for making new scientific findings and biologically plausible therapeutic discoveries.

Toward clinical translatability, we note that many of the drugs identified by comboFM model have been or are currently being explored in clinical settings against the specific cancer type, either as single agents or in combination with other drugs (see new Suppl. Table 5 for examples). We have highlighted selected combinations with high translatability potential in the manuscript, including ALK inhibitor combined with proteasome inhibitor, and EGFR inhibitor combined with chemotherapy lomustine in lymphoma cell line (SR), which carries ALK-fusion gene (pages 9-10). All of these four inhibitors have been studied in clinical settings extensively and their potential in treating lymphoma patients has been demonstrated. Similarly, comboFM predicted the well-known role of HDAC in melanoma, as most of the drug combinations predicted for BRAF-mutant melanoma cell line (MALME-3M) included HDAC inhibitor romidepsin as combination partner (see Suppl. Fig. 7, e.g., romidepsin-everolimus, romidepsin-vismodegib, romidepsin-cladribine, romidepsin-oxaliplatin, romidepsin-dactinomycin). Individually each of these inhibitors (or in combination with other drugs) have exhibited promising results either in preclinical or clinical settings against melanoma (page 11). These results already unveil the potential of our method for predicting combinations with translational potential, but these findings definitely warrant further validation in proper clinical trials.

Looking forward, we believe that once the model accuracy has been confirmed in the cell line resources, such as NCI-ALMANAC, where lots of training and testing data are available in multiple cancer types, we envision that the carefully-validated model will be applicable also to data from cancer patients. For selected cancer types, including leukemias and other hematological malignancies, there starts to be sufficient molecular and drug response profiling data available from patient-derived primary cells that can be used for training cancer type-specific prediction model, that will be applicable also to new patients with the same cancer type (please see refs. [30] and [31] in the manuscript). Once similar data from other cancer types becomes available, comboFM will enable also pan-cancer analyses, similar to the current analyses in the NCI-ALMANAC cell lines, by which one can also study similarities and differences in combination predictions between multiple tissue types, and even find new uses of combinations approved for one cancer type in other indications (drug repurposing). To investigate this further, we searched the ClinicalTrials database and found that many of the combinations predicted in the NCI-ALMANAC cell lines have already been tested in clinical trials (see new Suppl. Table 5). Interestingly, most of the combinations are tested in different indications than what was predicted based on the cell lines, suggesting that the same combinations might show therapeutic efficacy also in other cancer types. Even though these examples of clinical studies cannot obviously prove the clinical success of the in vitro synergy predictions, they indicate that the combinations have most likely already passed the required safety and tolerably testing in animal models and human subjects, as well as have shown efficacy in disease-relevant in vivo models.

We agree with the reviewer that translatability to patients is an important point to consider; however, we feel that proving the clinical success or translational impact of the in vitro synergies is a completely different scientific question that is independent of the comboFM method and its current cell line predictions. The pharmacodynamics is obviously very different between in vitro and in vivo systems, so even if the patient's tumor harbors the same mutation or other biomarker, it is not guaranteed that the patient would respond to the combination treatment due to additional genetic and non-genetic modifiers of treatment response. Therefore, we feel that the question of clinical translation is outside of the scope of the present work, where the aim was to develop an efficient method to predict biologically plausible and sample-specific drug combinations among the immeasurable number of possible drug combinations. However, the combinations with translational potential provide preliminary indication that our in vitro combination predictions can be predictive of therapeutic combination responses. Furthermore, we demonstrated here that the measured and predicted combination effects show rather heterogeneous profiles across the tested cell line panel, suggesting that patient-specific combination prediction and testing will become critical for therapy decision in clinical practice. Therefore, comboFM approach is expected to prove useful also for accelerating drug combination testing when applied to individual patients in the future, which will likely facilitate the community to select most effective cancer therapeutics for individual patients.

My recommendation would be a major rewrite and resubmission highlighting the 'significant jump' in translational impact of this work over just being a synergy predictive model. All minor and major comments and feedback, along with key references, have been added as highlights and comments to the submitted PDF (see attached).

Our response: We thank the Reviewer for detailed reading and commenting of the manuscript. Please find below our point-by-point responses to each of the comments marked in the PDF version.

Review #3 Detailed comments from the PDF

Abstract:

"We present comboFM, a machine learning framework for predicting the responses of drug combinations in pre-clinical studies, such as those based on cell lines or patient-derived cells." While I agree with the premise, preclinical response modelling has been done extensively. It is the translatability to patients that is the major bottleneck and need of the hours. The authors should not only highlight this fact, but also discuss at length the translatability credentials of comboFM.

Our response: We agree and have added further discussion of the translational aspects of the model in the revised version (pages 11-12).

Introduction

*“experimental screening of drug combinations quickly becomes impractical, as the number of conceivable drug-dose combinations increases rapidly with the number of drugs and doses.”
sentence needs simplification*

Our response: We thank the reviewer for pointing this out and have now simplified the sentence (page 2).

*“However, despite the potential value of such datasets, the high dimensionality of the underlying dose-response data and the inherent complexity of drug interaction patterns across various doses pose challenges to accurate modeling of drug combination effects.”
... as well as natural biological variability leading to lack of consistency of experimental outputs across the same cell line-treatment pairs.*

Our response: We thank the reviewer for pointing out the inconsistency issue that is indeed valid for all the preclinical experiments. We have now included this point in the limitation section of the revised section (page 12), where we argue that it is important to develop and initially evaluate the prediction models in large enough and standardized cell line resources, such as NCI-ALMANAC, to avoid the reproducibility issues in the development phase. However, we also think it is critical to test the model predictions later using different experimental setups in the same cell lines, like done here with the in-house testing, to show that the predictions are robust enough against such biological and technical variability. We feel this is highly important also when going to more translational studies, based on patient-derived ex-vivo responses, to guarantee the predictions are not due to just technical noise or biological variability.

Results

*“The first scenario of predicting new dose-response matrix entries corresponds to filling-in the gaps in partially measured dose-response matrices.”
how accurate is the method in filling up the ‘gaps’ in DR matrices?*

Our response: The accuracy of comboFM in filling in the gaps in the dose-response matrices is very high; the result is shown in Figure 2, together with the other prediction settings.

“random forest”

Minor comment - suggest using italics or upper case first letters to distinguish methods from usual text across the manuscript.

Our response: We thank the Reviewer for a nice suggestion. We have now used italics to distinguish the methods from the rest of the text.

*“the combination response in colon cancer appeared marginally more difficult to predict than the other tissue types, yet the the 5th order comboFM was still the most accurate method.”
This observation needs to be discussed further. The authors right point out earlier that data quality and quantity is a major limitation for such approaches. So, is this relatively tricky*

prediction for CRC lines a reflection of that, or is the biological space being modeled highly diverse?

Our response: The combination responses in colon cancer cell lines are only marginally more difficult to predict (please see Figure 3), which is likely explained by other variation in the data, as the number of colon cancer cell lines (6) is similar to the other cell lines and thus the data quantity is not a limitation here.

“As the ultimate interest often lies in discovering the most synergistic drug combinations” It actually is not. I do agree with the authors that predicting synergy was indeed the ‘ultimate interest’, but translatability of synergy scores to the clinic is non-existent. The critical component of any such predictive model is to comprehensively assess which of the synergies as ‘actionable’. The authors should describe these in detail, either by assessing which of the predictions have clinical relevance and applications, or by bringing in clinical genomics and drug response space into the model’s latent space. Please see Palmer and Sorger (2017).

<https://www.ncbi.nlm.nih.gov/pubmed/29245013>

Our response: We have now reworded this sentence to better capture the benefits of the comboFM model (page 6). The model enables accurate predictions of the full dose-response matrices between drug-dose combinations, not synergy of the combination per se. We chose to also test how accurately the comboFM-predicted dose-response matrices can predict the combination synergy, as this is one of the objectives of the drug combination experiments in the field. However, we totally agree with the reviewer that the synergy prediction is not the ‘ultimate interest’, rather providing the patients with effective combinatorial therapeutics, that often are needed to treat advanced cancers. We have also cited the interesting paper by Palmer and Sorger (new ref. [33]) that shows that while some cases of truly synergistic drug interactions provide also synergistic benefit, for most of the FDA-approved drug combinations, only little evidence of additivity or synergy was observed in preclinical models. This rather surprising result may also partly reflect the sub-optimal predictivity of the PDX animal data for clinical responses, and also the fact that historically combination clinical trials have been (and are still partly carried out) without any biomarkers for patient selection. We and others (see e.g. new ref. [34]) believe that patient stratification based on predictive markers will likely reduce variability in individual therapy responses in clinical trials, and contribute to achieving truly synergistic responses to combination treatments in future studies. comboFM and other models that use gene expression and other genomics data when making combination predictions are expected to identify also such predictive biomarkers for combination responses, once effective feature selection is implemented in the models.

“Importantly, the drug combination synergies can be accurately computed based on the predicted dose-response matrices also in the most challenging scenario of predicting new drug combinations,”

Good to see the predictive performance, what do the authors mean by ‘most challenging scenario’? Are these drugs the model has never seen? What is the distance (chemical and biological) between the training space to the independent validation space?

Our response: The “most challenging” referred to the third scenario of Figure 1, that of predicting new drug combinations that have not been measured in any cell line in the training data, however we assume the drugs have been individually measured, i.e. the monotherapy response is known. However, we now have reworded the sentence, without referring to “most challenging”, as indeed even more challenging scenarios can be constructed. The predictive performance of comboFM and compared methods are shown in Figure 2 in all three prediction scenarios. As comboFM learns from interpolating in the space of dose-response matrices through computation of latent factors representing similarly behaving drug combinations from the response tensor alone, the differences between these prediction scenarios come from the interaction terms that are available to the prediction model and not from the chemical and biological distances between the training and testing spaces. We have now added further discussion about this in the limitation section of the discussion (page 12).

*“172 unmeasured drug combinations across all the 60 cell lines, which resulted in a total of 10 320 predicted complete dose-response matrices.”
The authors should make the selection criteria for these 172 more obvious.*

Our response: We thank the reviewer for pointing this out. We have now clarified the selection of the combinations for the prediction and experimental validation on page 5.

*“highly synergistic effects only in a subset of all the cell lines and tissue types.”
why is this filter necessary? what defines ‘high synergy’? Is there a threshold that needs to be applied?*

Our response: Here, we considered drug combinations with a synergy score (NCI ComboScore) in the top 10% in a particular tissue type as highly synergistic combinations (as specified on page 6). Filtering based on highly synergistic effects in a subset of cell lines and tissue types was applied to demonstrate the ability of comboFM to perform in a more challenging task of predicting cancer-selective combinations, rather than broadly toxic combinations that kill most cancer cells, but which may also induce toxicities in the healthy cells.

*“16 KRAS-mutants pancreatic ductal adenocarcinoma cell lines (unpublished data).”
how similar are the assays with the data comboFM trains on?*

Our response: Both in-house and NCI-ALMANAC drug combination assays were carried out in 384-well plates and CellTiter-Glo reagent was used to measure the cell viability. The differences between two assays are (NCI-ALMANAC) was carried out in 3X3 combination matrices whereas in-house pancreatic cancer screening as well as experimental validation of the predicted

combinations were also carried out in 8X8 settings to obtain higher resolution. ALMANAC screens were carried out as 2 days end point assay whereas in-house screens were for 3 days, since the optimal effect of most of the drugs are seen in 72h. The ALMANAC assay lacked positive (cell killing) control and drug effects were quantified as growth percentage (with inclusion of time zero measurement) whereas in in-house assay we included both positive and negative control and drug effects were quantified as percent inhibition, normalized to both controls. This main difference between the assays is now mentioned on page 11.

“when considering positive Bliss score as evidence for a degree of synergy”

This is another key limitation of predicting synergy - even though from a large panel, different measures of synergy (Bliss, Loewe, HSA etc.) do show a high degree of correlation, at a smaller scale this agreement goes missing. The authors should comment on the choice of an independence model over an additive model.

Our response: As comboFM predicts the underlying dose-response matrices instead of predicting the synergies, one can compute any synergy score based on the predicted dose-response matrices. We assessed the synergies using four different and commonly used synergy scoring models (Bliss independence, Loewe additivity, zero interaction potency (ZIP) and the highest-single agent (HSA) models, see Supplementary Figures 6-7), which demonstrates the robustness of the comboFM predictions across various synergy models. We only included the Bliss independence score in the main Fig. 2 due to its popularity in drug combination studies and connection to the NCI ComboScore (and limited space in the main paper), but results with the other synergy models agree with those obtained with the Bliss model and provide further evidence of the synergy of these combinations. As noted earlier, synergy prediction was not the main point of the current study, and therefore we feel that more detailed comparison of the synergy models are outside of the scope of this work. We have added a citation to an excellent review article that provided careful and useful comparisons of similarities and differences of these and other synergy models in practical applications (new ref. [32]).

“in-house experiential validations,”

Do the authors mean experimental, or are referring to ‘expert’ validation?

Our response: Apologies for the typo, we meant to say “ in-house experimental validation“; it is now corrected in the manuscript (page 7).

“The prediction of high synergy between the first-generation inhibitors of ALK and proteasome for lymphoma cell lines highlights the potential of comboFM to predict biologically plausible combination effects. The comboFM model identified also another unique drug combinations effective against the SR cell line, the combination of EGFR inhibitor gefitinib with an approved chemotherapy lomustine for lymphoma treatment.” Biological rationale for both highlighted results driven by feature identification methods are needed to assess interpretability of the model.

Our response: The SR cell line harbors an ALK fusion, hence it is expected to be affected by ALK-inhibitors. Similarly, proteasome inhibitors have shown promising results against lymphoma. As most of the targeted single agents provide only short-lived therapeutic effects, because of resistance development, the biological rationale is that two even mildly effective inhibitors when used in combination are expected to enhance the effect and potentially overcome the resistance, leading to sometimes more durable clinical responses (page 9). Notably, comboFM made this prediction without knowledge of the ALK fusion status of the SR cell line, i.e. this biological rationale was not available for the model, rather the model mainly relies on the similarity of observed responses of similar drug-drug combinations, that is, interpolation in the dose-response matrix space, while the cell line and drug descriptors have an minor role in determining the prediction. We have clarified this aspect of the comboFM in discussion (page 12).

For the rationale of EGFR inhibitor, following text has been added to the manuscript (page 9):

“One of the mechanisms inducing resistance to ALK inhibitors is activation of EGFR, as they signal through similar downstream pathways. Brigatinib, a dual ALK/EGFR inhibitor, is therefore being explored in clinical settings (NCT01449461) against lymphoma and lung cancer patients. Our comboFM method predicted combination partners to extensively explored inhibitors (ALK and EGFR) for lymphoma, which we were able to also validate in the experimental setting.”

“Discussion”

The discussion section needs further details addressing choices, biological interpretation, and more importantly limitations of the methodology. While FMs have been gaining a lot of traction as discussed in this manuscript, there are a few limitations from being a very general model that requires a lot more explanation of the features capturing the signal and driving performance, to the fundamental publications for FM highlighting the need for clear optimisation when it comes to dealing with dense data leading to binary outcomes. A discussion of the gaps of this approach would be very valuable to the scientific community.

Our response: We thank the review for pointing this out. We have put comboFM forward mainly as an accurate tool for predicting dose-response values of drug combinations, which we see as a valuable tool for drug screening. We agree that there is a lot of work to be done in elucidating the predictions and understanding when and why certain drug combinations have synergistic responses, and other tools and approaches besides FMs will be needed to answer these questions. We have now added discussion about the biological interpretation of the results and highlighting these and other limitations of FMs (page 12).

Reviewers' Comments:

Reviewer #1:

Remarks to the Author:

I thank the authors for all the revisions done, which are mostly satisfactory. However, there are some important points that are still unclear.

MAJOR

After tuning one RF hyperparameter, the results of Figs 2 only show now a slight performance gain by ComboFM-5. Please update all other figures with RF results (FigS1, FigS4, FigS10...).

The authors have also tested the models on the non-overlapping test set I proposed (FigS3). However, in addition to the aggregated way the results are presented, the correlation with measured response and comboscores should be shown for each cell line (like in FigS2). The performance baseline, RF, is also missing: does comboFM perform better than the tuned RF on every cell line? The caption should clearly state how many monotherapy response values from the test set is comboFM using to predict test set comboscores and that RF uses none in comparison. The current conclusions in page 7 are in my opinion overoptimistic. Yes, 0.91 Rp is achieved on predicting test set response, but for comboscores this is only 0.4-0.5 Rp and still have to see the Rp per cell line (the most relevant scenario to identifying new drug combinations to test prospectively) and we have not seen yet the performance of RF without using test set monotherapies.

Regarding Point 9 in the response letter, it should have been stated in the manuscript that, while bliss scores obtained prospectively (FigS6) are around 10, the maximum in NCI-Almanac are around 500 (FigS3), which shows that the degree of synergy is low. The manuscript would benefit from explaining what it could be done to achieve higher synergies with machine learning modeling.

MINOR

The top legend in FigS3 states 'in other combinations', should this be 'in other training combinations'?

FigS1: reference should be measurements?

Reviewer #3:

Remarks to the Author:

Thanks to the authors for comprehensively addressing the issues raised as part of the first submission, wherever possible. I am now happy for this to be forwarded for publication.

Reviewer #1

I thank the authors for all the revisions done, which are mostly satisfactory. However, there are some important points that are still unclear.

After tuning one RF hyperparameter, the results of Figs 2 only show now a slight performance gain by ComboFM-5. Please update all other figures with RF results (FigS1, FigS4, FigS10...).

Our response:

The aforementioned figures in the previously-revised version already included the updated RF results after tuning the RF hyperparameter. For instance, FigS4 in the first revision version shows significantly better performance of comboFM-5 in recovering the ComboScores (Pearson correlation of 0.72 in the new drug combination prediction setting), compared to RF model (Pearson correlation of 0.49 in the same setting). Similarly, the performance of the two methods across the NCI-60 cell lines was already included in the revised Figure 3, but we have now also added new Supplementary Figures 3 and 4 showing the predictive performance of comboFM-5 and random forest in each of the cell lines, in terms of the both measured response and ComboScore, further demonstrating that comboFM-5 generally outperforms RF across all the cell lines and tissue types ($p < 0.00001$ for paired comparisons of the improvement with a Wilcoxon signed-rank test across the cell lines in each prediction setting).

The authors have also tested the models on the non-overlapping test set I proposed (FigS3). However, in addition to the aggregated way the results are presented, the correlation with measured response and comboscores should be shown for each cell line (like in FigS2). The performance baseline, RF, is also missing: does comboFM perform better than the tuned RF on every cell line?

Our response:

We thank the reviewer for the suggestion. We have now added results on the non-overlapping validation set also for Random Forest (new Supp. Fig. 5). We have also added new Supp. Fig. 6 showing the correlations with measured responses and ComboScores in each cell line separately for this non-overlapping set. These results show that 5th order comboFM clearly outperforms RF in each cell line in terms of correlations with both measured responses and ComboScores ($p < 0.00001$, Wilcoxon signed-rank test). Notably, RF leads in some cases even to negative correlations with the ComboScores.

The caption should clearly state how many monotherapy response values from the test set is comboFM using to predict test set comboscores and that RF uses none in comparison.

Our response:

There seems to be a misunderstanding here. To make a fair comparison, RF uses exactly the same monotherapy responses as comboFM in these results. The data used for training and testing the different methods is always the same in order to provide a fair comparison.

Regarding the caption, we have now added the number of monotherapy dose-responses from the validation set used in the training (476).

The current conclusions in page 7 are in my opinion overoptimistic. Yes, 0.91 Rp is achieved on predicting test set response, but for comboscores this is only 0.4-0.5 Rp and still have to see the Rp per cell line (the most relevant scenario to identifying new drug combinations to test prospectively) and we have not seen yet the performance of RF without using test set monotherapies.

Our response:

We would like to point out that this performance is for the non-overlapping validation set and the aforementioned Pearson correlations are for the combinations where neither drug has been observed in any combination or cell line in the training set. This is a setup that many of the existing prediction models cannot even deal with, as also demonstrated by the added RF results, which in some cases even lead to negative correlations with the ComboScores.

As noted in the limitation section of the revised discussion (page 12), comboFM is designed to interpolate in the space of dose-response matrices through the computation of latent factors representing similarly behaving drug combinations from the response tensor alone (this is analogous to recommender systems that are grouping users by the movies they have liked in the past), while the drug and cell line descriptors merely fine-tune the predictions. Hence, comboFM obtains the best predictive accuracy when the training data contains information on how the individual drugs behave either alone or as part of other combinations. In the above type of scenarios, where one would like to perform predictions for molecules with no prior combination response data in any cell line, the computed latent factors are not as helpful, which results in weaker correlations compared to the other type of prediction settings presented in Figure 1. However, in our opinion, Pearson correlation above 0.9 for the prediction of responses also under this challenging setting with no prior combination response data available is already impressive.

We believe our work demonstrates how data-driven prediction models, such as comboFM, can provide accurate predictions of the most potent combinations in practical scenarios, providing faster discoveries compared to the current, mostly hypothesis-based selection of drug combinations for further development. We also demonstrated the accuracy of comboFM predictions by experimentally validating some of the predicted synergistic combinations in our subsequent laboratory experiments, further indicating the biological relevance and robustness of the predictions also to experimental factors.

Regarding Point 9 in the response letter, it should have been stated in the manuscript that, while bliss scores obtained prospectively (FigS6) are around 10, the maximum in NCI-Almanac are around 500 (FigS3), which shows that the degree of synergy is low. The manuscript would benefit from explaining what it could be done to achieve higher synergies with machine learning modeling.

Our response:

The Bliss scores and ComboScores have different units and thus the magnitudes of these scores cannot be directly compared (please also see our response in the previous revision

round related to this comment below). To illustrate this point, we have prepared the below figure showing the distributions of Bliss scores in the in-house background dataset and ComboScores in NCI-ALMANAC, with extreme percentiles indicating synergy (or antagonism) marked with dashed lines. For instance, the 99th percentile of the Bliss scores is around 10, whereas the corresponding 99th percentile of ComboScores is around 150. This comparison shows that the two scores follow in general similar background distribution, while their ranges are drastically different, where only the most extreme scores indicate true synergy (or antagonism).

We believe our work demonstrates how advanced prediction methods, such as comboFM, are required to achieve improved prediction of synergistic combinations, especially in more challenging prediction scenarios. This is expected to lead also to higher synergies of the experimentally tested combinations, as one can focus in the follow-up analyses only on the most potent combinations among the massive number of combinations in the original space.

Our response from previous revision round:

The 16 selected combinations from comboFM predictions shown in Figure 4 were experimentally tested in-house using a different assay than that in NCI-ALMANAC, with the aim to assess how accurately the comboFM predictions can be validated also using different laboratory setup, compared to the NCI-ALMANAC training data, to prove its robustness to the experimental factors. In particular, in the in-house FIMM assay, the drug combination responses were measured in the form of percentage inhibition instead of percentage growth that is used in the NCI-ALMANAC assay. Hence, as the calculation of NCI ComboScore would require the combination responses to be measured in the form of percentage growth of the cell lines, computing NCI ComboScores for this experimental validation set was not possible. Therefore, we evaluated the combination synergies using four different and commonly used synergy scoring models (Bliss independence, Loewe additivity, zero interaction potency (ZIP) and the highest-single agent (HSA) models, see Supplementary Figures 6-7), which demonstrated the robustness of the comboFM predictions across various synergy models and experimental assays. We only included the Bliss independence score in the main Fig. 2, due to its popularity in drug combination studies and connection to the NCI ComboScore (and limited space in the main paper), but results with the other synergy models agree with those obtained with the Bliss model and provide further evidence of the synergy of these experimentally validated combinations. We feel this experimental validation is important for showing the robust performance of the comboFM, in addition to the computational validations in the NCI-ALMANAC data, and therefore we would like to keep these results even if NCI ComboScore could not be used.

MINOR

The top legend in FigS3 states 'in other combinations', should this be 'in other training combinations'?

Our response:

We thank the reviewer for pointing this out, and we have now reworded the legend as suggested.

FigS1: reference should be measurements?

Our response:

We thank the reviewer for the suggestion. In FigS1, the reference indeed represents measurements and we have now renamed "reference" as "measurements".

Reviewer #3 (Remarks to the Author):

Thanks to the authors for comprehensively addressing the issues raised as part of the first submission, wherever possible. I am now happy for this to be forwarded for publication.

Our response: We thank the referee for the positive evaluation of our revised work.